# Economic Inclusion: Green Finance and the SDGs

Arno J. van Niekerk 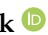

Department of Economics and Finance, Faculty of Economics and Management Sciences,
University of the Free State, Bloemfontein 9300, South Africa; niekerka@ufs.ac.za; Tel.: +27-51-401-3271

**Abstract:** Persistent economic exclusion and the high levels of natural resource depletion are alarming. The Sustainable Development Goals (SDGs) are among a few global initiatives aimed at bringing a turnaround in both of these areas of concern. Giving action to productive economic inclusion and transitioning towards a circular, regenerative economy is challenging for countries, particularly because of a lack of economic incentives. Green finance has emerged in the last few decades as a valuable mechanism that has the potential to meet this challenge. In answering the question of how to facilitate the necessary transition to a green, inclusive economy, the paper attempts to bring green finance and economic inclusion together as a possible means (like a bridge) to address economic exclusion and resource degeneration. That is the primary aim of the study, and it is investigated through an analysis of theoretical literature. The key findings include: a strong synergy exists between green finance and economic inclusion; different forms of green finance are able to facilitate economic inclusion; and green finance can be instrumental in attracting investors to fast-track SDG attainment. A key conclusion is that green finance can play a vital role in activating and prolonging broad-based benefit sharing in an eco-conscious way.

**Keywords:** economic inclusion; green finance; SDGs; circular economy; green investment

## 1. Introduction

Arguably, the key challenge of the world economy in the 21st century is balancing environmental concerns with socioeconomic needs and economic growth and development to ensure genuine economic progress for all. Fundamental obstacles that need to be overcome include: economic inequality (in all its variants); resource depletion (the net deficit); global imbalances and growing geopolitical tension between the East and the West; and, as time progresses, "technology unemployment" (or automation-induced unemployment). Economic sustainability remains an ideal that the world is still unable to reach. The Sustainable Development Goals (SDGs), in following the Millennium Development Goals (MDGs), are perhaps the closest we have come to establishing a set of tangible shared objectives—a relatively clear pathway—for economies around the world. The 2020–2022 COVID-19 pandemic, however, put everything on hold as far as sustainable economic progress is concerned [1]. In fact, the recession that most economies went through probably postponed the attainment of the SDGs by a further decade, from 2030 to 2040.

The question now is what are the key avenues to be followed that can fast-track progress towards the SDGs? One such avenue is green finance [2]. Another is prioritizing economic inclusion at all levels of the economy. A kind of "economic blueprint" needs to be developed on how green financing can unlock new economic potential in economies that leads to efficient economic inclusion, which results in economic sustainability. Since the reality of the 21st century world is very much integrated and interdependent, such a "blueprint" or "pattern" will need to carry the same characteristics in shaping what could potentially be a new economy. The struggle for the poor has always been to take part in creating value in the economy (either through employment or entrepreneurship), while the struggle for investors has always been creating a sustainable economy so that more can benefit. By combining green finance and economic inclusion, a bridge can be built between

these extremes to facilitate stable and effective flows that generate sustainable progress in a natural/unforced way. The paper thus examines the feasibility of building such a "bridge". How green finance may contribute to multiple facets of economic inclusion and the SDGs will be investigated in view of shaping a better integrated framework for the economy to ensure sustainable progress. The study's primary discovery is the significantly positive contribution that green finance can make to economic inclusion and, through their intersection, address the two main lacking requirements for a sustainable economy: proper environmental stewardship and improved economic equality. The paper is structured in a way to first provide clarity on key concepts and relevant theories. Second, it specifies the methodology and design used. Third, it examines the relationship between green finance and economic inclusion. Fourth, it explores the contribution that green finance can make to SDG attainment in the context of economic inclusion. Finally, before the conclusion, a discussion of the results and findings of the study follows.

## 2. Literature Review: Definitions and Theoretical Framework

In the absence of an official or commonly accepted definition of green finance, scholars, financial institutions, governments, and organizations working on/with green finance generally agree on the following characteristics: it refers to financial products, services, and investments that support and promote environmentally sustainable and socially responsible practices; it involves loans and investments that support activities with a positive environmental impact, including the acquisition of ecologically responsible products and services or the development of eco-friendly infrastructure and green technology (Green technology is defined as "technology whose use is intended to mitigate or reverse the effects of human activity on the environment" [3] (p. 1). Also known as eco-technology, it refers to the development and application of innovative and sustainable solutions that address environmental and ecological challenges.); and the financing of not only private but public green investments, such as financing public policies aimed at promoting the implementation of projects and initiatives that provide environmental protection and/or reduce ecological damage (e.g., reduce pollution) [2,4–7]. Zadek and Flynn draw a distinction between green finance and green investment in that the latter is a subset of the former that focuses more on allocating capital to projects, assets, or businesses that have a positive environmental or social impact, i.e., investing in sustainability [8]. Green finance is wider in scope, encompassing all financial activities, products, and services that support initiatives that are environmentally sustainable and socially responsible, plus it includes the operational costs of green investments, such as project preparation and land acquisition costs.

Another significant distinction is between climate finance and green finance. Climate finance—as a component of green finance—is particularly focused on addressing the impacts of climate change, whether through adaptation strategies or measures to mitigate and limit greenhouse gases (GHG). It also "...aims at reducing vulnerability of, and maintaining and increasing the resilience of, human and ecological systems to negative climate change impacts" [9] (p. 5). To do this, as part of green investments, climate finance employs market-based environmental policy instruments to enhance the ecological impact of investment strategies, thus trying to mitigate the adverse effects of climate change through pricing and trading mechanisms. In this way, for instance, it guides businesses in reallocating resources to improve investment sustainability while maintaining profitability [10]. Furthermore, green finance is often used interchangeably with sustainable finance. In this instance, green finance forms a part of sustainable finance, with the latter taking a broader view, encompassing not only environmental considerations but also social and governance factors to align financial activities with sustainability objectives. An example of this is ESG (Environment, Social, and Governance) as a framework now increasingly used to evaluate and measure the sustainability and ethical impact of a company or investment. According to Hayes, "ESG scores are a measure of how well a company addresses risks and concerns related to environmental, social, and corporate governance issues in its day-to-day

operations" [11] (p. 1). Within this framework, green finance functions as an instrument that redirects financial resources towards ecologically sustainable projects and investments, thus facilitating the transition to a low-carbon economy [12]. Through green bonds, green loans, sustainability-linked loans, impact investments, and insurance products with a sustainability focus, green finance focuses on the overall financial system and how it can be aligned with sustainability goals, such as the SDGs. Although all these concepts are related, there are mainly three clear facets of green finance that stand out, according to Lindenberg [4]:

- the financing of public and private green investments in areas such as water management (e.g., dams) and the protection of biodiversity (e.g., in landscapes), or in the prevention, minimization, and compensation of ecological damages (e.g., energy efficiency or infrastructure). Such financing includes project preparation costs and capital costs;
- the financing of public green policies that promote the implementation of environmental and ecological damage mitigation or adaptation projects and initiatives (and their operational costs). This may include measures like feed-in tariffs designed to incentivize the use of renewable energy sources; and
- contributing to the development of a green financial system that encourage, for instance, green investments (e.g., the Green Climate Fund or financial instruments for green investments like green bonds and structured green funds) and their specific legal, economic, and institutional framework conditions.

When considering the core components of an inclusive economy, it is clear that green finance forms an integral part of it. Such an economy is driven by inclusive growth with the aim of yielding genuine economic progress to promote equality of opportunity and broader well-being in terms of both people and the planet [13]. An inclusive economy is characterized by a commitment to ensure that opportunities for prosperity are widely shared among all segments of society and economic inequalities are reduced, so as to improve the well-being of especially those who are marginalized or disadvantaged. It is distinctive in the fact that it promotes the inclusion of all members of society in the growth process itself instead of distributing wealth among them only after periods of steep growth, which is usually the case with economic development [14]. Both green finance and an inclusive economy are strongly focused on improving economic sustainability from a systemic perspective. Apart from inclusive growth, four other main pillars of sustainability that an inclusive economy rest on include [13]:

- genuine economic progress: measuring progress beyond GDP per capita; that is, measuring collective well-being (Collective well-being involves sustainable and equitable opportunities, access, benefit-sharing, and balancing the preservation of human capital, ecological capital, and shared social norms in an innovative way [15].) by taking environmental and social costs into consideration to ensure net progress, thus enhancing the ecological yield (the harvestable population growth of an ecosystem) and social yield (the harvestable human capital development (net value added) in a community);
- a circular economy: unlike the linear economy of "take-make-use-waste", this economic model follows a different pattern to minimize waste through product-level reuse, such as repair or refurbishment; component-level reuse, such as manufacturing; and material-level reuse, such as recycling or upcycling in an environmentally-friendly way.
- a collaborative economy: instead of only relying on intermediaries/companies, consumers rely on each other in a peer-to-peer marketplace, sharing and having access to resources, goods, and services directly from each other, often through digital/online platforms. Also called a sharing economy, the emphasis is on temporary use (access) rather than ownership (e.g., Uber and Airbnb);
- inclusive policies and institutions: policies aimed at stimulating and generating inclusive growth, employment, inclusive development, equal access to opportunities, and

equitable distribution of benefits and resources; improved redistribution of income (fiscal policy), and a more inclusive financial system (monetary policy). Inclusive institutions create the framework (the legal, social, and economic structures, rules, and practices) for a just and accessible economic environment that promote economic inclusion and reduce disparities.

Key theoretical frameworks that green finance and an inclusive economy lean on are, firstly, ecological economics and environmental economics. Both study the interrelationship between the economy and the environment in the context of sustainability. Ecological economics treats the economy as a subsystem of the Earth's larger ecosystem, stressing the preservation of natural capital [16]. It is a transdisciplinary field that studies the interdependence and co-evolution of human economies (social dimensions of economic systems) and natural ecosystems. Placing strong emphasis on ensuring social justice and maintaining the integrity of ecological systems, the primary goal of ecological economics is to develop sustainable and equitable economic systems that respect ecological limits and promote human well-being [17]. It rejects the idea that man-made (physical) capital can be a substitute for natural capital. The concept of "ecosystem services" is highlighted and needs to be protected and properly maintained. These are the benefits that humans derive from ecosystems, such as clean air, clean water, and pollination. Being very concerned about the well-being of future generations and the sustainability of economic activities over time, ecological economics relies not on money as the primary decision-making factor but explores the role of ethics and values in decision making.

Environmental economics analyzes the costs and benefits of environmental policies, market-based instruments, and regulatory approaches to address environmental challenges. It often focuses on specific environmental problems and how economic solutions/tools can be used to address them. According to Chen, "the basic theory underpinning environmental economics is that environmental amenities (or environmental goods) have economic value and there are costs to economic growth that are not accounted for in more traditional models" [18] (p. 1). Using standard economic methods, such as cost–benefit analysis, it assesses the trade-offs between economic development and environmental protection. The primary goal of environmental economics is to find cost-effective solutions to environmental problems in the short-term and, as a counterpart, to internalize ecological externalities by quantifying the economic value of environmental goods and services (including access to clean water, clean air, forests (for carbon sequestration), the survival of wildlife, and biodiversity) to society [19]. Environmental economics provides the basis for pricing carbon emissions, creating market-based instruments such as carbon markets (e.g., buying and selling carbon credits), and promoting the efficient allocation of resources for environmental protection.

What these theories have in common is their emphasis on "regenerative economics". This is largely encapsulated in the vision for a green economy. Essentially, economics is defined as the study of how resources are distributed/managed. Conventional economic thinking orbits around economic growth. If the economy is growing, there is progress. With the new emphasis on the green economy, the accent is shifting from quantity (of economic activity, i.e., money changing hands) to quality of (human) life. It restores a much-needed balance in the economy. As Figure 1 shows, the formal economy is seen in the broader context of social and environmental resources where human capital and natural capital are balanced with financial capital (and not seen as externalities).

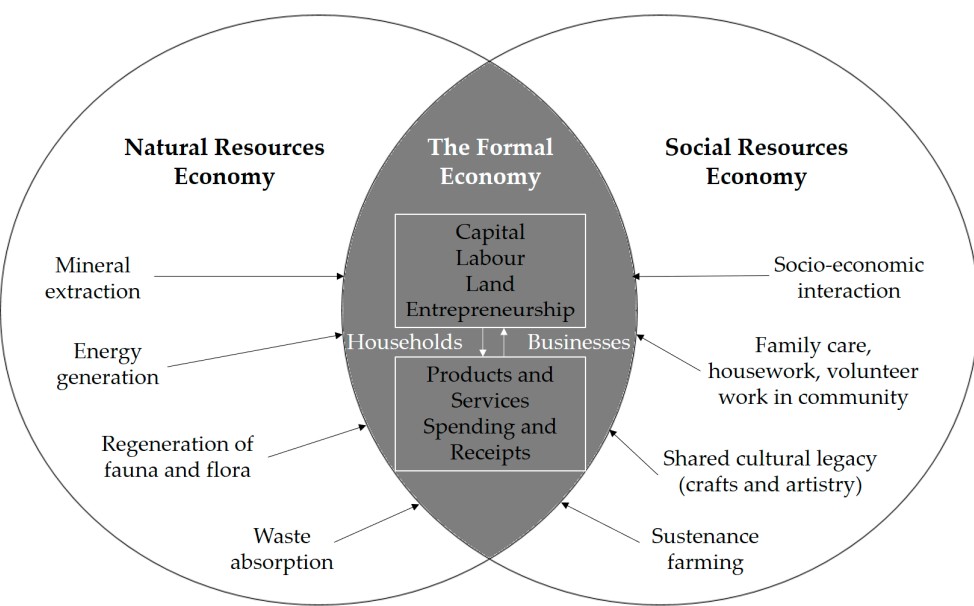

**Figure 1.** From a circular flow to a circular economy [20].

Appreciating the regenerative nature of the green economy sensitizes us to a deeper understanding of the regenerative capacity of the planet, our only basic resource. This leads to refraining from depleting non-renewable resources more rapidly than renewable alternatives can be developed and to constraining our waste production, including pollution, to a point where it remains within the planet's sustainable capacity. To be green, an economy must be both fair and efficient. In this context, the United Nations states, "a green economy is defined as low carbon, resource efficient and socially inclusive" [21] (p. 1). Hence, fairness and efficiency find expression in creating harmonious economic interactions between humans and nature. A constant process of regeneration (regenerative economics) works to regenerate capital assets, which include all forms of capital. All capital assets provide goods and services that are needed for, or contribute to, our well-being. What therefore sets the green economy apart from standard economic theory is the recognition that natural capital and ecological services hold economic value, and the implementation of a thorough cost accounting system that correctly assigns the costs imposed on society through ecosystems to the accountable party, whether they cause harm or fail to maintain an asset [22]. Growth in employment and income are driven by public and private investment into regenerative economic activities.

The green economy offers a macroeconomic strategy for achieving inclusive growth, placing a primary emphasis on investments, job creation, and skill development [20]. This will also improve resource efficiency, leading to increased well-being while reducing resource consumption and emissions. Central to advancing a green economy is access to green finance, technology, and investments [21]. Macroeconomic policy can play a vital role in facilitating a country's transition to a green economy through various tools and practices to build the necessary capacity and to mainstream eco-efficient production and responsible consumption behaviors. Critical to this are partnerships. Multi-stakeholder partnerships are needed to accelerate and consolidate sustainable changes in both consumption and production patterns. Collaboration between governments, non-profit organizations, international institutions like the UN, and the private sector is vital for promoting resource efficiency in a green economy. For the greening of an economy, each role player needs to do their part and work towards common goals, for example:

- states need to address regulatory frameworks; set standards for emissions, pollution control, resource management, sustainability practices; and build institutional capacity;

- businesses need to do the technical innovation for regenerative business practices; ensure a greening of their supply chain; and position themselves as responsible corporate citizens;
- the civil society can advocate for sustainability; live environment-friendly lifestyles; hold states and businesses accountable; provide training and resources, enabling communities to benefit from green initiatives like renewable energy projects or sustainable agriculture; and ensure equitable access to green opportunities (e.g., recycling projects); and
- international institutions can provide expertise, resources, guidance, and coordination at the global level to assist local economies in making the transition to a more balanced model.

What strengthens these practical outcomes and the collaboration that occurs spontaneously is the fact that it is anchored in new economic thinking, i.e., a coherent new theoretical framework that views balancing social, economic, and environmental factors as the basis for genuine progress. The last theory to round off the framework for the context at hand is called the "inclusive economic theory". This theory synthesizes rational economic thinking (realist neoclassical theory) with altruistic welfare- and utility-enhancing behaviors (neo-realist economic theory), which is equally evident in the economy. This provides a cooperative framework where consumers maximize utility and producers maximize profits as part of the shared collective ideal of wealth creation that benefits all, which is what true economic progress is about [23]. Given the market's inability to regulate itself and mankind's moral imperfectability, it points to the need for collective accountability in our economic system. Added to this, research has shown that where norms/values are more constantly applied, societies usually perform better [24].

An example of this is the Japanese economic system that see sacrifices as well-being-enhancing because group values (family, society, company, and government) are embraced [25]. In such a culture, people's collective well-being keeps improving in a setting where institutions create the space for building a consensus. This is not a rejection of markets; rather, it reorients them—through incorporating shared values—to the higher cause of communal satisficing (satisficing means one is seeking an acceptable solution, not the optimal one per se because it might be unattainable). In this way, well-being-enhancing opportunities are included for societies and individuals in the neo-realist paradigm as a counterpart to the enhancement of Pareto optimal welfare. In 1776, Bentham already concluded that societies where collective utility, not just individual utility, is maximized performed better [26]. This holds even truer in today's interdependent world where the need for more realism and wider inclusion in economic theory has become essential. This inclusion, which starts to transform modern capitalism (reversing private interests over public welfare), comprises both biocentric ethics and shared community values. As a custodian of this, inclusive economic theory—as part of new economic thinking—satisfactorily meets the main task of economic theory: to correctly decipher complexities and formulate appropriate and reality-based interventions.

A new development that bears significance to both green finance and economic inclusion is Central Bank Digital Currencies (CBDCs). In view of the tremendous growth of cryptocurrencies since 2009, central banks started to develop their own version, backed by central banks themselves, called CBDCs [27]. According to the International Monetary Fund (IMF), by the end of 2023 over 100 countries (60% of the world) were exploring and developing CBDCs [28]. As a digital currency that must be universally acceptable and able to be exchanged by peers, the main motivations behind CBDCs include:

- Financial inclusion: this is especially the case for low-income countries that view CBDCs as the ideal mechanism to involve the 1.4 billion people (globally) still outside the formal banking system [29].
- Policy control: they help to improve control over monetary policy and macroeconomic policy, i.e., changing the structure of money demand by speeding up the circulation of currency, making central bank reserves more controllable and money supply more

intelligent (with more direct control over interest rates), and can increase the volatility and expand the currency multiplier to improve the transmission effect of existing policy tools [30].

- Reducing illicit transactions: by implementing CBDCs through deploying digital currency on a permission distributed ledger technology or blockchain, central banks can reduce the use of other means of payment (e.g., cash) related to illicit activities. All transactions can be traced, hence each sender and receiver can be uniquely identified. Financial fraud in the circular economy can be detected and eradicated in this way [31].

- Green and sustainable finance: in line with growing concern for the environment, CBDCs can play a vital role in transforming the infrastructure of finance to become more sustainable. In a study by Yang et al., the results show that in China's case, CBDCs significantly promote the issuing of green bonds, especially in the manufacturing sector and in state-owned enterprises [32]. Other green benefits include reducing $SO_2$ (sulfur dioxide) emissions, NOx (nitrogen oxide) emissions, and improving the green land ratio essential to sustainable development.

In the scientific community, an ongoing discussion and apprehension persist regarding the possible environmental consequences of virtual currencies. From a sustainability point of view, it is important to note that while CBDCs are not cryptocurrencies, energy use is still required for their operation [27]. The energy sources employed in cryptocurrency mining can lead to heightened pollution, with particular concern regarding air pollution. Leslie adds to the discourse by highlighting the increase in electronic waste and the extra energy demands to offset the heat generated by these platforms [33]. CBDCs should therefore not be unaware of the social and environmental costs related to decentralized finance. For this reason, the Bank for International Settlements (BIS) has made the promotion of green and sustainable finance, including CBDCs, central to its objectives from 2022 onward [34]. To this end, it is worthwhile to also link the development of CBDCs with the SDGs through combining financial inclusion and green finance. CBDCs could thus be a key role player, especially in the context of SDG number eight: "Promote sustained, inclusive and sustainable economic growth, full and productive employment and decent work for all" [35]. This emphasis is supported by Maltais and Nykvist, who found in their study on green bonds in Sweden that a burgeoning green bond market is positively influencing individuals' dedication to sustainability [36].

Several theories have surfaced to elucidate the rationale behind employing instruments such as CBDCs to improve financial sustainability from both a social and environmental perspective. The public good theory of financial inclusion argues that CBDCs should be used as a tool to broaden financial inclusion as a public good that benefits all citizens, leaving no one out [37]. The government might need to provide subsidies for the provision of financial services through CBDC payments, aiming to incentivize a broader adoption of CBDC accounts for enhanced financial inclusion. Another theory is the dissatisfaction theory of financial inclusion. When bank clients, for instance, lose trust in or become frustrated by their institution's financial service delivery, it may result in them exiting the financial system. A central bank can leverage such public discontent by offering an alternative means—such as a CBDC—to access the financial system without the need for direct interaction with a commercial bank [38]. An example of this is the eNaira issued by the Central Bank of Nigeria in October 2021 as a legal tender.

Current empirical research concentrates on examining the advantages and use cases of CBDCs in both developing and developed nations. In Nigeria's case, the eNaira is leading to the digitization of value chains in Nigeria; it is enlarging Nigeria's growing digital economy; and offers lower financial transaction costs, thus improving accessibility [37]. In CBDC feasibility studies, Maniff found that CBDCs may not automatically lead to increased financial inclusion and/or ecological sustainability if its design is in conflict with the other objectives for creating the CBDC [39]. It is also vital that CBDCs be complemented by new, more efficient technologies that provide supplementary features as part of modernizing

payment systems for greater inclusion. The ability of CBDCs to offer a practical remedy for the challenges in cross-border payments is highly valued.

In an empirical study by Murakami, Shchapov, and Viswanath-Natraj, the researchers found a number of unresolved issues in the design of CBDCs [40]. They used a two-agent framework to illustrate that CBDCs can enhance financial inclusion predominantly when households utilize them as a means of saving to stabilize consumption. They also discovered that retail CBDCs are more valuable and advantageous in economies with limited financial inclusion. In a calibrated overlapping-generations (OLG) model developed by Banet and Lebeau that quantifies the financial inclusion–intermediation trade-off, it was found that "the channel through which a CBDC could impact inclusion depends on its usage cost relative to that of bank deposit accounts" [41]. Although the former does not have to be inherently less expensive than the latter to enhance inclusion, the authors demonstrate that CBDC designs featuring lower usage costs (and reduced interest rates) create a more favorable trade-off between inclusion and intermediation. This trade-off involves finding a balance between expanding access to financial services for underserved populations and the potential risk of reducing the role of traditional financial intermediaries. While the quantitative analyses by Banet and Lebeau are focused on the United States, the same mechanisms could translate directly to emerging markets where access to a bank account is not so widespread and where financial inclusion's impact on poverty is arguably greater. For future research, measuring the potential trade-off between inclusion and intermediation or disintermediation in these countries may be particularly significant.

Lastly, in a study by Rahman et al., in which the authors investigated the number of studies in green finance and sustainable development, they found that much fewer of these studies have been done in developing countries [42]. This research gap points to the need for more comparative analyses of the interaction between finance and environmental sustainability between developed and developing countries. It will also help policymakers to better contextualize the policies they implement to establish complementary frameworks between public and private sectors in this regard. For instance, there are a number of associated variables that should be taken into account when adopting and implementing green finance, such as managerial variables (monitoring and evaluation); banking sector sophistication (e.g., technology use); secondary environmental effects (e.g., additional energy uses due to technology, electronic waste, new buildings/infrastructure); legal factors; and additional costs related to corporate social responsibility [43]. The role of the Green Climate Fund (GCF) is crucial for developing countries in order to successfully implement sustainable green finance. As part of the UN Framework Convention on Climate Change, and mainly funded by leading industrialized countries (given them being the main culprits in climate deterioration), the GCF can help fund efforts to reduce greenhouse gas emissions and to stimulate green finance in developing countries [44]. It could also support central banks in formulating green finance policies that allocate climate funds. Having studied the role of central banks in green finance extensively, Gunningham found that the significance of central banks in executing and monitoring green finance activities can be proportionately more impactful in developing countries [45]. In investigating, by means of a meta-analysis, studies that consolidate, analyze, and identify trends in green finance, Rahman et al. found that annual publications in green finance have risen gradually between 2014 and 2022 [42]. This indicates growing scholarly interest, which would benefit practitioners, policymakers, and service providers—especially in the banking and financial sectors. An aspect that requires specific attention is substantial risks associated with green finance solutions being taken into account when banks make decisions [46]. The lack of clear steps to decrease green finance risks is a research gap, requiring the development of robust risk mitigation models. How to promote more responsible financial behavior from a societal and environmental perspective while reducing risks related to green finance is a major subject of inquiry. Overall, sustainable finance certainly plays a vital role in achieving the SDGs.

## 3. Methods and Materials

This study makes use of a theoretical literature review analysis to explore, first, the synergy potential of green finance and economic inclusion, and secondly, how they as a combination can fast-track SDG attainment in pursuit of improved economic sustainability. Such an analysis comprises examining, evaluating, and synthesizing the existing theoretical frameworks, concepts, and applications relevant to these two research focus areas (or hypotheses). It aims to provide a comprehensive overview and qualitative analysis of the existing literature. This allows for a coherent interpretive and narrative-based approach to gaining insight, emphasizing the qualitative analysis and synthesis of information rather than a quantitative or systematic analysis that employs formal statistical methods.

This approach is considered suitable for answering the paper's research question: what incentives can be created through green finance and economic inclusion to facilitate a transition to a sustainable, regenerative economy exemplified by the SDGs? A specific focus is placed on how such a combination can bridge or fill the gap between the extremes in the economy by creating incentives/avenues for the poor to be included in the economy and incentives for investors to create additional capacity in the economy that accelerate economic inclusion. Criteria or pillars of economic inclusion are used as a basis (the scope) for the synthesis and evaluation in this descriptive theoretical review analysis. The relevant available literature was used in the process, comprising various sources such as databases, scholarly journals, books, and other credible sources.

In terms of the inclusion criteria employed in the study, the relevant studies selected comprised peer-reviewed journals, technical literature in books, and scholarly works at the United Nations, the World Bank, the European Commission, and the OECD (Organization for Economic Co-operation and Development)—all focused on the intersection between the environment, finance, and economic inclusion: that is, sustainability. The publication years of these studies are predominantly post-2000, and more specifically after 2014, when green finance became more mainstream and formalized as a distinct field, and after the SDGs were developed in the build-up to the Paris Agreement in 2015. Some studies before 2000 are also featured, taking cognizance of the fact that environmental regulations started in the 1970s with the first Earth Day in 1970 and, especially since the 1980s, when socially responsible investing (SRI) started to lay the groundwork for integrating environmental considerations into investment decisions. Abstracts, conference papers, reviews, commentaries, and preprints were not considered for this study because they are not published in peer-reviewed journals. There might be some sampling bias given that this review study only includes articles published in the English language.

With regards to previous studies that investigated the same area of research using a similar methodology, the author has not found one, which gives substance to the contribution of this study. Other studies that deal with similar topics are restricted to only focusing on green finance, financial inclusion, and sustainability but not on the broader concept of economic inclusion as outlined in this paper or as underscored in the SDGs. The emphasis on economic inclusion and giving more clarity on what it means in combination with green finance in the context of the SDGs is what arguably sets this study apart. Hence, the paper does not replicate or adapt existing methodological approaches.

The reason why the study does not follow a quantitative approach are the limited assessment data currently available with regards to economic inclusion, green finance, and measuring SDG attainment—especially in a cross-country sense for comparative purposes. Some measures of economic inclusivity do exist (e.g., the McKinley Inclusive Growth Index and the Global Database of Shared Prosperity), but they are not yet comprehensive enough as they need further development for them to be fully useful, particularly when different contexts (e.g., developing and developed countries) are compared. While progress has been made to measure the extent to which countries are achieving the SDGs, such as the UN working with national statistical offices, there are still many concerns about (1) the gaps in the required data; (2) difficulties in disaggregating statistics to reveal trends in specific subpopulations (for example, the poor, urban versus rural populations, and

persons with disabilities); and (3) sufficient capabilities still need to be properly developed because measuring and monitoring SDG progress is a complex, multifaceted process involving actors at the subnational, national, regional, and global levels. The choice of methodology in this study stems from, first, the need of a conceptual analysis and the contextual development of a theoretical framework for the combination of green finance, economic inclusion, and the SDGs. Secondly, given the novelty of these concepts in the financial sector and the growing interest they attract (and the complexity involved), a proper review of the literature would help plot them on the ideological/conceptual spectrum in economic theory. Thirdly, a qualitative approach enables the study to provide practical examples and applications of these concepts in the financial industry, as recorded in the literature.

The pillars or criteria of economic inclusion employed in the study come from an in-depth study by this author in writing the book "The Inclusive Economy: Criteria, Principles and Ubuntu", in 2022 [13]. From over a thousand academic sources, the following five criteria of an inclusive economy were identified: inclusive growth; genuine economic progress; collaborative economy; circular economy; and inclusive policies and institutions. Of course, there were other aspects of an inclusive economy that were highlighted apart from these five criteria, but they were the ones most often pointed to in the literature, which comprised peer-reviewed journals, books by leading economists, and research by international institutions such as the World Bank, IMF, WTO, UN, OECD, and others.

The literature search conducted in this study first involved a search for keywords such as economic inclusion, financial inclusion, green finance, SDGs, circular economy, green bonds, and green investment. Secondly, sources were selected on the basis of how they examined the interaction between specifically green finance, economic inclusion, and the SDGs. It was important to not just find out what the literature says about each of these subject areas individually, but how researchers view the connection between them. Many of these scholarly works emphasized different combinations of these three main focus-areas, as well as different subcomponents of the three areas (e.g., financial inclusion as a component of economic inclusion), but none took cognizance of their synergy as comprehensively as this paper does.

## 4. Green Finance as a Means of Economic Inclusion

Green finance is one of the key interventions for constructing a more inclusive and sustainable economy. History has shown, especially since the first industrial revolution, that finance can be a powerful enabler of human progress [47]. Considering the synergy between the concepts and the fact that they are of mutual benefit, green finance clearly has the capacity to contribute to economic inclusion. Here are some of the most evident ways [2]:

- employment creation: by supporting investments in renewable energy, energy efficiency, sustainable agriculture, and other environmentally friendly sectors, green finance can create new job opportunities, specifically for those in marginalized groups (e.g., entry-level), and create entrepreneurship prospects for innovators and small business owners;
- access to clean energy: by improving living standards, green finance can facilitate access to clean and affordable energy sources for underserved communities. This may also enhance economic opportunities by enabling small businesses to operate more efficiently [48];
- financial inclusion: by providing access to financial services, such as affordable loans and microcredit for individuals and businesses engaged in green activities, green finance can combine financial inclusion with green investment by enabling them to invest in sustainable projects and grow their businesses—even at a small scale;
- community development: by building green infrastructure and revitalizing urban areas, and improving access to clean water and sanitation, green projects enhance living

conditions in local communities and create economic opportunities (e.g., upcycling projects);

- sustainable agriculture: apart from increasing yields and income in rural communities, access to green finance, knowledge, and technology for sustainable farming methods can also promote food security. This will empower and benefit small-scale farmers immensely;
- access to green technologies: funding can be provided through green finance to resource-poor households for adopting green technologies, such as solar panels and energy-efficient appliances. Access to eco-technologies can reduce energy costs, improve the well-being of households, and may even involve them in innovative solutions for ecological challenges;
- access to green markets: by supporting the integration of disadvantaged communities into green supply chains and markets (e.g., renewable energy, regenerative agriculture, eco-tourism, and eco-friendly products), green finance can broaden economic opportunities for producers in remote or impoverished regions. For example, plants with medicinal value known in those areas can unlock latent economic potential. Green finance can provide funding for research laboratories and starting businesses that benefit whole communities.

Green finance can certainly be instrumental in driving economic inclusion as a catalyst/facilitator of investments in sustainable and environmentally responsible activities. By addressing ecological challenges and promoting equitable access to opportunities, green finance is invaluable to genuine economic progress based on inclusivity. More specifically in the context of the core components of an inclusive economy, green finance firstly promotes inclusive growth. This is done through different ways in which green finance is able to contribute to various components of inclusive growth. For instance, in the case of pro-poor growth—as one facet of inclusive growth—green finance needs to be able to reduce inequality in either relative terms (poor people's income increases relative to that of the rich) or absolute terms (fewer people ending up below the poverty line). Key requirements are that it must be disadvantage-reducing and lead to an increase in benefit-sharing [49]. This means that a distributional shift is necessary, and more of the poor need to be included in income-generating processes.

In this context, economic growth that brings changes to the distribution of income is the kind of (inclusive) growth to be pursued. Therefore, the question is, which types of green finance contribute to such growth? Since green bonds, as a type of fixed-income instrument, are specifically earmarked to raise money for climate and environmental projects, it will be necessary for those projects to include and empower the poor through, for instance, sustainable agriculture and other energy-efficient projects in sectors such as manufacturing, construction, and transportation to increase their income relative to that of the rich. Other types of green finance such as green funds and green stocks (equity financing) have the potential, through directing investments to companies that prioritize ESG criteria or to green mutual funds, to enable disadvantaged communities they are partnering with to share in the benefits of the returns on investment, which would also reduce the disadvantages experienced in these communities. Impact investing is another type of green finance that can address skewed income distribution experienced by the poor by involving them directly in investments that generate positive environmental and social impacts alongside financial returns. These investments can take the form of equity investments in social enterprises, microfinance, or investments in clean technology companies. Microfinance institutions providing microcredit can partner with other institutions that provide green loans to enable individuals and small businesses to access funding for green projects, such as solar panel installations or eco-friendly farming. In fact, as highlighted by the Alliance for Financial Inclusion, financial institutions in general can—as a commitment to inclusive green finance—provide crucial assistance to those navigating an unpredictable environment by promoting green products across various areas, including savings, credit, insurance, money transfers, and innovative digital delivery channels [50].

Another facet of inclusive growth is broad-based growth. How can green finance be instrumental in stimulating this kind of growth? The goal of broad-based growth is to involve more of the poor and marginalized in the process of growth through productive employment [51]. Also called employment-intensive growth, it means that at any given level of growth, the economy needs to become more labor-absorbing. Specific areas that are integral to broad-based growth where green finance can make a contribution include increased participation (access to productive opportunities for micro-entrepreneurs, small business owners, smallholders, and members of cooperatives, and giving them more market access) and increased sustainability (financing that values the natural resource base, causes the workforce to become healthier, better educated, and more inclusive, generates technology appropriate to local needs, and enables greater involvement of individuals and communities in the decisions that affect their well-being). Green finance initiatives that can assist with this include:

- job creation through renewable energy projects—in collaboration with disenfranchised communities—where investors directly finance, through community-based green funds, renewable energy projects, such as solar, wind, hydro, and geothermal energy installations (and giving such groups special benefits through power purchase/cost-free agreements);
- green bonds and green loans for projects tailored to the employment needs of marginalized communities and green funds for human capacity building by supporting educational and training programs focused on sustainable practices, equipping individuals with the skills needed to participate in green industries; and
- involving community members in employment-intensive green investment projects, such as: environmental infrastructure (e.g., water treatment facilities, waste management systems, and public transportation); carbon markets—financed through carbon market instruments—that entail the buying and selling of carbon credits and green derivatives; projects focused on conserving natural resources and responsible land use; and investing in green technology startups and then marketing green technologies, such as clean energy solutions, sustainable materials, and eco-friendly products.

The third facet of inclusive growth where green finance can make a contribution is shared growth. The goal of shared growth is to ensure that the fruits of growth are shared in a manner that reduces income inequality considerably and eradicates poverty [52]. The emphasis is on the redistributive aspect, but in the sense of economic growth occurring first, then the subsequent allocation of its benefits. The primary focus is on the importance of continuity: from growth to sharing. High and sustainable growth must create equal access (social inclusion) to improve people's living standards as they become more productive. Shared growth literally means "growing together". Green finance can play a key role in unlocking the fuller impact of economic growth to communities in a sustainable manner. For example, new growth opportunities in the renewable energy sector (and in the whole climate solutions industry), financed by green bond grant schemes, green venture capital, and carbon finance, can involve marginalized communities in new ventures, such as green real estate (e.g., sustainable building projects, green renovations, and energy-efficient real estate developments); clean transportation (e.g., electric vehicles and infrastructure for cycling and walking); and upgrading energy-efficient facilities to reduce energy consumption and lower GHG emissions. This even leaves those communities themselves in a better condition in terms of sustainability, emphasizing the opportunity to grow together.

A key factor for the growth of green finance, as Wang and Wang point out, is to vigorously develop green financial projects and products so as to better involve green commercial enterprises, and to properly facilitate their green transition [53]. Thomas highlights further that "decisive action and a rapid pivot towards the opportunities presented by the zero-carbon economy [is vital] . . . to achieve the scale of the transformation required. Cross-sector collaboration and the focused application of the creativity, innovation and skills of the financial services industry to finance the global transition will" [54] (p. 1). The significant challenge at the moment—and what limits the impact of green finance in

creating an inclusive economy—is the green financing gap. Green investment in different sectors has experienced rapid growth in recent years, notably in areas such as green buildings, clean energy, and renewable energy production. However, despite this surge, a large financial gap still exists to de-carbonize the economy.

According to a report from the United Nations Environment Program, addressing climate change, biodiversity, and land degradation targets necessitates closing a substantial 4.1 trillion USD financing gap by 2050 [55]. Investments in nature-based solutions were 133 billion USD in 2022, with the majority sourced from public funds. The reasons for this financing gap, according to Desalegn and Tangl, are inadequate funding, suboptimal green project selection and management, the trade-off between risk and return, and a deficiency in analytical tools and expertise for identifying and evaluating green project risks [56]. Regulatory hurdles, in particular, have been identified as the primary obstacle to advancing green finance and greening the financial system. To fill the green financing gap, Jeffrey Sachs and others made the following suggestions [47]:

- focus on green banks' continued development: to offer improved credit terms for clean energy initiatives, be more capable of combining smaller projects to attain a commercially appealing scale, enhanced ability to do green financial product innovation, and expanding their markets by disseminating information about the advantages of clean energy [42];
- green central banking: being responsible for financial and macroeconomic stability, central banks ought to be fully involved in addressing climate-related and other environmental risks at a systemic level [27]. Helping to develop green finance models and, at a policy level, address environmental risk and promote sustainable finance;
- non-bank financial institutions: institutional investors such as pension funds and insurance companies are well-positioned to direct corporate capital allocation towards more sustainable endeavors. They hold long-term financial resources that are suitable for investing in green infrastructure and are the largest suppliers of capital to listed companies.
- new financial technologies (FinTech, Hong Kong, China) can unlock new opportunities in green finance by leveraging blockchain applications for sustainable development. This includes blockchain use cases for renewable energy, decentralized electricity markets, carbon credits, and climate finance, as well as innovating financial instruments such as green bonds; and
- more incentives for inclusive growth to accelerate collective change: in this way, investors and disadvantaged communities can take co-responsibility together for the environment and collaborate on solutions of mutual benefit (higher rates of return and income growth).

Green finance also complements another core component of an inclusive economy, transitioning to a circular economy. In a circular economy, products and services are traded in closed loops or "cycles", involving the reuse and recycling of material goods to eliminate waste and using more renewable resources. It is an economy where growth is not driven by or reliant on finite resources; in fact, it progressively disassociates (decouples) growth from finite resource consumption [57]. Such a sustainable closed-loop economic system functions in a way that all waste is either utilized as regenerative resources for nature, recovered resources, or by-products (i.e., food) for another production process—thus creating a self-sustaining cycle. Economic growth in a closed-loop system is called "green growth". Green growth aims to maintain the circulation of resources for as long as feasible, while also ensuring a sustained level of product quality for users. It is inherently regenerative, designed to preserve the maximum value of parts, products, and materials. According to the OECD, "green growth means fostering economic growth and development while ensuring that natural assets continue to provide the resources and environmental services on which our well-being relies" [58] (p. 1). Hence, green growth strategies try to address a twin challenge: expanding economic opportunities for everyone given a growing global population; and dealing with environmental anxieties that, if neglected, could hinder our

capacity to capitalize on these opportunities. Using circularity principles, green growth is distributive by design, meaning, for instance, that economic growth is desirable as long as it creates employment.

Green growth is intricately connected to the principles of a green economy and low-carbon or sustainable development. A key catalyst for green growth is the shift towards sustainable energy systems. By applying inclusive business models to achieve green growth, environmentally sustainable agriculture can, for example, be promoted, preserving biodiversity and ecosystem services. Supported by the effective implementation of green policies, job opportunities can be created, particularly in sectors like renewable energy, sustainable forestry, and green agriculture. Since the connection between green growth and green finance is obvious, the question is how green finance creates opportunities for the advancement of green growth. It provides financial support to circular economic and environmentally sustainable growth initiatives in several ways:

- funding renewable energy projects: green finance channels funds into renewable energy projects, such as solar and wind farms, facilitating the transition to clean and sustainable energy sources. Typical green finance instruments used are renewable energy certificates, green loans, climate funds, green venture funds, and tax credits and subsidies;
- supporting energy efficiency initiatives and circular business models: financial institutions offering green finance can support projects aimed at improving energy efficiency in industries, buildings, and transportation—contributing to overall sustainability—and fund products and materials designed for longevity, reparability, and, if possible, recycling;
- promoting sustainable agriculture: green finance can, through agricultural sustainability bonds and impact investing, fund agricultural practices that prioritize environmental sustainability, promoting the use of eco-friendly farming techniques and technologies;
- facilitating green infrastructure development: investments in green infrastructure, such as eco-friendly transportation and sustainable urban planning, are made possible through green finance, fostering sustainable development and circular resource management;
- encouraging corporate sustainability: in the form of green equity, green revolving credit facilities, and corporate social responsibility funding, green finance directs capital towards eco-friendly businesses, encouraging broader corporate adoption of circular practices across industries, involving more stakeholders and role players in the supply chain;
- issuing green bonds: probably the most commonly used instrument in green finance is green bonds, which are earmarked to fund almost all projects with environmental benefits, particularly those contributing to green growth and circular practices;
- incentivizing sustainable investments and green innovation: by promoting resource efficiency and the development and implementation of innovative technologies and processes, green finance facilitates investments through offering financial incentives, such as preferential interest rates or tax benefits, into circular solutions and eco-innovations;
- aligning investments with ESG criteria: Environmental, Social, and Governance (ESG) criteria guide green finance, ensuring that investments foster long-term green growth (e.g., in eco-friendly production, using sustainable production methods that apply circularity);
- funding circularity at the design phase: innovation in product design that expands recyclability and upcyclability (upcycling means transforming waste materials or unwanted products through reuse, upgrades and repairs, into new materials or products of higher quality or value than the original) are ideal for receiving financial backing through green finance.

Green/circular finance creates opportunities for green growth by directing capital towards projects and initiatives that prioritize resource efficiency, waste reduction, and using renewable materials and contribute to the overall well-being of the planet, including all its people and resources. This financial approach encourages businesses to adopt circular practices such as recycling, reusing, and reducing the environmental impact. Supported by green policies from the government and a green financial system, a more integrated framework is created that is essential to an inclusive economy. When all parties are collaborating towards the same sustainable goals, as in the case of the SDGs, a synchronized harmony can filter down to involve everyone in building a balanced new economy. This should be the new "trickle-down effect". A vital stimulus for such collaboration is financial inclusion (financial inclusion means providing greater access to reasonably priced financial products and services for poor and low-income individuals, as well as businesses with limited resources, to meet their needs and help them build wealth). The financial sector has a key role to play, given their central role in the economy.

Green finance presents a number of mechanisms and strategies to accelerate financial inclusion, as part of promoting overall economic inclusivity. The focus of green finance in this context is also to strengthen the role of financial inclusion in assisting vulnerable communities in developing resilience and minimizing the impacts of climate change-induced losses. A key challenge, on the other hand, is findings by studies that in some cases there is a positive correlation between financial inclusion and increase in energy consumption, resulting in more emissions in a region [59–61]. Hence, green finance presents an opportunity to address this by not just directing financial resources to underserved individuals and businesses, but to allocate funds to them in a way that it enables them to utilize the resources they gain access to in an eco-friendly way (e.g., green consumption and production). This creates a virtuous circle. As Wang et al. explains, "Inclusive finance provides credit to environmentally sustainable projects which are devoted to meeting consumers' green consumption needs; and conversely consumers' green purchasing behaviour reinforces the growth of these environmentally sustainable projects, leading to a green consumption mode for inclusive growth" [62] (p. 248). For small and medium-sized businesses, inclusive finance helps them gain an advantage from access to credit, resulting in job creation, innovation, and growth. More specific ways in which green finance promotes financial inclusion include:

- community-level projects: green finance can target projects that benefit underserved populations. This may include initiatives related to clean energy, water management, and sustainable agriculture in areas with limited access to traditional financial services;
- green financial products for the underserved: green bonds or sustainable investment funds can be designed to be inclusive, giving access to a broad range of investors, including those with smaller financial resources, to contribute to and benefit from eco-friendly projects. This will also address the inclusive development and financial needs of marginalized communities sustainably, ensuring a reciprocal sharing of the financial benefits generated;
- capacity building: through green finance initiatives, financial literacy training and can be provided by financial institutions to empower individuals and businesses to make informed decisions about sustainable investments and participate in green economic activities [63];
- microfinance for green enterprises: microfinance institutions funding green enterprises to encourage entrepreneurship and economic empowerment at the grassroots level;
- financial inclusion promoting green economic efficiency: studies showed that the four main types of inclusive financial services (payments, savings, credit, and insurance) have a notably positive impact on green economic efficiency [62,64]. The transition to mobile payments is significant. Credit constraints on high-polluting firms play a strong role in accelerating their process of industrial transformation and green innovation;
- leveraging FinTech to enhance financial inclusion: digital platforms can provide access to green financial products and services to reach a wider audience, including those in

remote areas [65]. Utilizing blockchain makes green finance transactions more transparent, reducing the risk of fraud and ensuring that financial services are accessible to a broader population. A study by Liu et al. found that by streamlining financial processes, increased automation and improved data analytics through FinTech can reduce both energy usage and operational costs, making green community banking more affordable and accessible [66];

- furthering the SDGs: according to Mabeba, financial inclusion has been recognized as a catalyst for seven of the 17 SDGs [67]. Being aligned with broader inclusivity goals such as the SDGs, green finance becomes a tool for achieving integrated development outcomes, making eco-friendly investments accessible to a diverse range of stakeholders. Such trends are similar in developed and developing countries [68]

## 5. Green Finance and the SDGs

In the last few decades, there has been a significant rise in global agreements about addressing environmental concerns, especially those caused by the economy. After the SDGs came into being in 2015, the Paris Climate Accords—covering climate change mitigation, adaptation, and finance—were adopted by 195 countries in 2015, and the European Green Deal was signed in 2020 as a set of policy initiatives with the aim of making the EU climate neutral (a net zero emitter) by 2050 [69]. The World Economic Forum (WEF) also had a Green Horizon Summit to mobilize green capital in 2020. In 2021, at the UN COP26 conference, "new building blocks were put in place to advance implementation of the Paris Agreement through actions that can get the world on a more sustainable, low-carbon pathway forward" [70] (p. 1). Involving almost 200 countries, the negotiations resulted in the Glasgow Climate Pact. The UN conference underscored the significance of green bonds, highlighting that a mere 1% of the roughly 300 trillion EUR in financial assets within the markets would be sufficient to realize the SDGs [71]. While the global green bond market has grown significantly from 93 billion USD in 2016 to almost 900 billion USD in 2023, this is still only around 14% of total global bond issuance [72]. It certainly is a market with strong growth potential. The US, China, and France are leading the way as the largest issuers of sustainability-linked bonds.

Of global significance is the role green finance can play in helping build broad momentum in achieving the SDGs, as part of furthering economic inclusion (also emphasizing the role of women) [73]. The push by global role players to bring a transition in investment strategies towards green outcomes is enabling the implementation of some goals in a general sense, such as SDG 1 (No poverty), SDG 2 (Zero hunger), SDG 3 (Good health and well-being), and SDG 10 (Reduced inequalities) [56]. More specific contributions of green finance to the SDGs in the context of economic inclusion, comprise:

- SDG 7—Affordable and clean energy: investments in renewable energy projects, making clean and sustainable energy sources more accessible and affordable;
- SDG 8—Promote sustained, inclusive, and sustainable economic growth, full and productive employment and decent work for all;
- SDG 9—Industry, innovation, and infrastructure: investments in green technologies and sustainable infrastructure that enable circular industrial development methods and cost-reduction through innovation;
- SDG 11—Sustainable cities and communities: funding eco-friendly urban development, resilient infrastructure, and sustainable transportation to promote inclusive cities;
- SDG 12—Responsible consumption and production: green finance fosters sustainable and inclusive business practices, distributed manufacturing, and regenerative consumption;
- SDG 13—Climate action: financing projects that mitigate climate change, reduce carbon emissions, promote green economic efficiency, and that respect limits (ecological ceiling);
- SDG 14—Life below water: investments in sustainable fisheries, marine conservation, and reduction of ocean and river pollution contribute to cultivating a healthy marine life;

- SDG 15—Life on land: providing finance for projects focused on biodiversity conservation, sustainable land management, circular production, and combating deforestation;
- SDG 17—Partnerships for the goals: financial institutions, states, businesses, and civil society collaborating to promote green finance and nurture communitarian networking.

## 6. Results Discussion

When considering the feasibility of green finance and economic inclusion to, as a combination, unlock not only new economic potential for the inclusion of marginalized groups but also new opportunities for investors to explore, this study is fairly unique in terms of placing strong emphasis on the potential impact of this combination. Studies in sustainable development that generally focus on these areas of research often lack clarity on what economic inclusion entails and lack comprehension of the variety of green finance methods available and the full potential they hold of being purposeful instruments of economic inclusion. The paper's uniqueness and the novelty it introduces is further demonstrated by its emphasis on synergy between the incentives on both sides of the economic spectrum: investors see green investments and finance not only as new avenues through which to generate higher returns but also as a way to expand productive capacity in underserved communities that can stimulate economic growth; and the poor value such investments not just because of the income potential through new job creation, but also because it tangibly improve their living conditions and quality of life. Adding the fact that such incentives on both sides create natural momentum towards achieving the SDGs further distinguishes the paper compared to previous studies. The results from the study underscore the positive potential of the synergy between green finance and economic inclusion in attaining the SDGs.

The immediate political implications of green finance and economic inclusion—and especially their synergy potential—is evident in how it can influence policy making. For instance, it gives impetus for governments to give even higher priority to policies that support sustainable development, environmental protection, and social equity. Among others, this should lead to the adoption of legislation and policies aimed at promoting renewable energy, sustainable infrastructure, and inclusive economic growth. Green and inclusive policies should set the tone for economic development. Monetary policy is another area where financial inclusion can now become an even stronger focus, especially in combination with green finance. In the context of national elections, voter preferences may shift as awareness of environmental issues and social disparities grows, influencing electoral outcomes. With the SDGs in mind, the findings of this study may also lead to increased global cooperation as governments engage in international forums and agreements to address climate change, priorities for sustainable development, and addressing economic inequality.

Lastly, the limitations of the study include limited comparative data availability (especially on economic inclusion) and hence comprehensive statistical analysis and results; methodological constraints due to the emphasis on a theoretical analysis; and complexities with regards to the interdisciplinary nature of the study (environmental, social, and economic), especially when applied to environmental policies, financial mechanisms/instruments, and social conditions/dynamics. These aspects provide opportunities for future research, especially with regards to a large-scale statistical analysis where different contexts within and between countries can be compared. Furthermore, the increased use of FinTech to enhance economic inclusion and promote green finance are fertile areas to be explored.

## 7. Conclusions

E.F. Schumacher warned that "It is inherent in the methodology of economics to ignore man's dependence on the natural world" [74] (p. 73). This was further underlined by Manfred Max-Neef, who demonstrated that "When macroeconomic systems expand beyond a certain size, the additional benefits of growth are exceeded by the attendant costs,"

meaning that the quality of life decreases as the economy grows [75] (p. 116). He was referring to his "threshold hypothesis" and identified balancing factors such as social equity and ecological parity as answers to this dilemma in the economy. Conversely, if the economy keeps exceeding the threshold, the dual threats of financial exclusion and ecological deficit are risks that can become growing sources of financial instability. As the article identified, green finance can be a highly significant instrument, or bridge, to facilitate economic flows that include the poor in creating economic value and enable investors to help create a sustainable economy that includes broad-based benefit-sharing for longer, eco-consciously.

Constructing an inclusive economy does require a level of commitment by all economic role players that arguably has never been seen on a global scale. In the 21st century, the global community shares challenges (socioeconomic and environmental) that can only be addressed as a collective. In view of economic globalization (i.e., the integration of economies) and the fact that the world is moving into the Fourth Industrial Revolution (4IR), despite increasing geopolitical tensions between the East and the West, efficient financing methods to build bridges become critical. Green finance is central to this. There are, however, important limits and challenges to overcome with regards to green finance. Firstly, for investors and corporations there is uncertainty about the analysis of green finance, including the lack of consistency in assessing corporate greenness, the unclear definition of corporate greenness, and the lack of reliable and standardized environmental data to do proper comparative assessments [76]. Secondly, legitimacy issues with green finance involve questions about the green credentials of certain green bonds. This relates to both the projects that are funded and the sustainability credentials of the issuers. Firms that were involved in such green bond controversies include Vigeo Eiris in 2016 and Repsol in 2017 [77]. Another legitimacy issue is the tendency of companies to overstate or falsely claim environmental benefits, called greenwashing. The lack of clear criteria and verification processes contribute to this, but of even greater concern is the apparent lack of regulatory oversight and the inherent capital arbitrage opportunity presented to issuers [78].

Thirdly, there are microeconomic challenges to green finance: information asymmetry problems; problems with internalizing environmental externalities; maturity mismatches between short-term and long-term green investments; and deficient coordination between financial and environmental policies [2]. Fourthly, according to Ntsama et al., in low-income and middle-income countries, underdeveloped green markets can be attributed to institutional, financial, technical, and political barriers [79]. Not just in these countries but often also in higher-income countries, it is a problem that inadequate institutional design within the renewable energy sector results in a misaligned incentive structure for its participants [80]. Fifthly, there are challenges with technology such as inefficient or inappropriate technologies and infrastructure (e.g., software risk pertaining to blockchain-based security tokens to address green market failures). All these challenges represent opportunities for further research. More studies should specially be conducted on how to enhance green finance for green investment to affect green growth more deliberately.

The synergies between green finance, economic inclusion, and the SDGs are evident. The SDGs function like a roadmap towards a new economic reality, the inclusive economy functions like a destination—which the SDGs lead to—while green finance function like a bridge that unlocks the financial flows along the way to reach the destination. Green finance can fulfill a coalescing role in bringing collaboration between many different partners across sectors, industries, and geographies. As a common denominator, it can create connections between actors to work together in synchronized purpose, building financing bridges across boundaries. In this way, an integrated network can be established that opens the way to a new economy. According to John Fullerton, "A regenerative economy seeks to balance: efficiency and resilience; collaboration and competition; diversity and coherence; and small, medium, and large organisations and needs" [81] (p. 38). To establish a better integrated framework for the economy, intentional economic inclusion

is fundamental. In this process, green finance and a green financial system are vital mechanisms for generating and channeling the financial resources to replenish our natural resource base and to accelerate human resource development.

**Funding:** This research received no external funding.

**Data Availability Statement:** The data presented in this study are available on request from the corresponding author.

**Conflicts of Interest:** The author declares no conflicts of interest.

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
