# Peer review of "Economic Inclusion: Green Finance and the SDGs"

_sustainability, doi:10.3390/su16031128_

Round 1

Reviewer 1 Report

Comments and Suggestions for Authors

The paper reviews the literature to explore how green financing can contribute to economic inclusion to encourage sustainable development. The topic is highly relevant, and the manuscript is well written. I appreciate Section 2 (Definitions…). The authors have done a great job in defining at the outset all key terms to avoid any confusion later on.

Nevertheless, I do have some serious concerns regarding the novelty and methodology of the work. When I read the ‘gap’ in lines 46-51, I got excited because there was so much generic information out there in the world. In the article, I was hoping to learn something new and valuable in a more structured way. However, the paper lacks a methodology and does not show how the authors conducted the literature review. Overall, despite a promising knowledge gap, the paper without a methodology seems an opinion piece and only covers the generic information that most readers would already be aware of. I would suggest adding a section on methodology to strengthen the quality and value of the paper.

Author Response

Thank you very much for taking the time to review this manuscript. Please find the detailed responses below and the corresponding revisions and corrections highlighted in the manuscript that was re-submitted to the editor of the journal.

  1. As per your request, a full section on methodology ("Methods and Materials") were added to the paper in lines 274-293. It definitely adds value to the paper.
  2. Other adjustments were also made by other reviewers to improve the paper. They are all highlighted in yellow.

Kind regards,

Reviewer 2 Report

Comments and Suggestions for Authors

I am appreciative of the chance to read this essay. Thank you for sharing your interesting findings. This article complies with all relevant main requirements. 

The goal is evident, and prior research that is referenced and utilized as examples supports the background. Analyzing the impact of the Covid period would also be interesting, too.

In light of the existing corpus of knowledge about the matter, the writers ought to formulate and highlight a few research hypotheses. 

They fundamented very well the entire theory, but the article is too general. It looks like a literature review.

Do they have access to any data source (statistical data) in the fields? Some interesting statistical analysis might be developed.

The limitations of the study have to be emphasized.

Overall, the article is consistent within itself.

Author Response

Thank you for taking the time to review this manuscript. Please find the detailed responses below and the corresponding revisions and corrections highlighted in the paper that was re-submitted to the editor of the journal.

  1. In view of your concern that the article might be too general and could benefit from a few research hypotheses, a full section on methodology ("Methods and Materials") were added to the paper in line 274-293. I am sure it will help to bring clarity.
  2. With regards to your question on statistical data, my access is restricted a bit, but the study is not specifically focused on that. I am aware that it could add value, however the idea was to do the study firstly from a theoretical perspective.
  3. I have added a section ("Results Discussion") before the Conclusion in response to your request for identifying the limitations of the study. I have also added other valuable aspects in that section.
  4. Note that I have also made other adjustments requested by other reviewers to improve the paper. They are all highlighted in yellow.

Kind regards,

Reviewer 3 Report

Comments and Suggestions for Authors

Review: The article addresses persistent economic exclusion and high levels of natural resource depletion as alarming concerns. It focuses on the Sustainable Development Goals (SDGs) as global initiatives to tackle these issues. It highlights the difficulty of achieving productive economic inclusion and a circular economy due to a lack of economic incentives. The paper suggests that green finance can be a valuable tool to address these challenges, creating synergies with economic inclusion and attracting investors to expedite SDG attainment. It concludes that green finance plays a vital role in activating and prolonging broad-based benefit-sharing in an eco-conscious manner.

The article explores the timely and contemporary topic of green finance, sustainability, and how green finance can contribute to achieving the Sustainable Development Goals (SDGs). It delves into the novel intersection of economic inclusion, resource conservation, and the pivotal role green finance plays in advancing the SDGs.

The author should enhance the introduction by explicitly stating the primary discovery made in this study. Additionally, it is advisable to conclude the introduction by offering an overview of the paper's structure, outlining the key components that will unfold the interconnected dynamics of green finance, economic inclusion, and sustainable development.

The second section is excessively broad and would benefit from being labeled as a "Literature Review." Following the literature review, the author should make further divisions, starting with a "Materials and Methods" section. This should encompass the materials analyzed and provide a clear presentation of the methodology employed. This restructuring is both essential and imperative for enhancing the clarity and organization of the paper.

Furthermore, the literature review conducted is somewhat vague and could benefit from a more comprehensive approach. It is recommended to include additional relevant references in this field. Some suggested references for enhancement include: [List of suggested references]. This will contribute to a more robust and well-grounded literature review, strengthening the foundation of the study:

World Bank. (2007). Finance for all? Policies and pitfalls in expanding access. The World Bank.

Demirguc-Kunt, A., Klapper, L., Singer, D., & Ansar, S. (2018). The Global Findex Database 2017: Measuring financial inclusion and the fintech revolution. World Bank Publications.

Náñez Alonso, S. L., Jorge-Vazquez, J., Echarte Fernández, M. Á., Kolegowicz, K., & Szymla, W. (2022). Financial Exclusion in Rural and Urban Contexts in Poland: A Threat to Achieving SDG Eight?. Land, 11(4), 539.

Kara, A., Zhou, H., & Zhou, Y. (2021). Achieving the United Nations' sustainable development goals through financial inclusion: A systematic literature review of access to finance across the globe. International Review of Financial Analysis, 77, 101833.

Arner, D. W., Buckley, R. P., Zetzsche, D. A., & Veidt, R. (2020). Sustainability, FinTech and financial inclusion. European Business Organization Law Review, 21, 7-35.

Náñez Alonso, S. L., Jorge-Vazquez, J., Reier Forradellas, R. F., & Ahijado Dochado, E. (2022). Solutions to financial exclusion in rural and depopulated areas: Evidence Based in Castilla y León (Spain). Land, 11(1), 74.

Alonso, M. P., Gargallo, P., López-Escolano, C., Miguel, J., & Salvador, M. (2023). Financial exclusion, depopulation, and ageing: An analysis based on panel data. Journal of Rural Studies, 103, 103105.

Tay, L. Y., Tai, H. T., & Tan, G. S. (2022). Digital financial inclusion: A gateway to sustainable development. Heliyon.

Kuada, J. (2019). Financial inclusion and the sustainable development goals. In Extending financial inclusion in Africa (pp. 259-277). Academic Press.

Nguyen, N. P., & Mogaji, E. (2021). Financial inclusion for women in the informal economy: An SDG agenda post pandemic. Gendered perspectives on Covid-19 recovery in Africa: Towards sustainable development, 213-236.

It is absolutely essential to include a dedicated "Methodology" section that encompasses the materials and methods employed by the author in conducting the study. This section will provide a transparent and detailed account of the author's approach, enhancing the overall rigor and clarity of the research.

Prior to the conclusions, it is crucial to include a section labeled "Results Discussion." In this section, the author should:

  1. Conduct a comparative analysis of their results with findings from previous studies.
  2. Clearly articulate the novelty introduced by their article.
  3. Discuss the political implications stemming from their research.
  4. Address any limitations encountered during the study and outline potential avenues for future research.

This comprehensive results discussion will enrich the overall impact and scholarly contribution of the paper.

Author Response

Thank you very much for taking the time to review this manuscript. Please find the detailed responses below and the corresponding revisions and corrections highlighted in the paper that was re-submitted to the editor of the journal.

  1. As per your request, I have enhanced the Introduction by explicitly stating the primary discovery made in the study. This is highlighted in lines 52-55.
  2. A component was also added at the end of the Introduction to be an overview of the paper's structure. This is highlighted in lines 55-61.
  3. Responding to your request, I have relabeled the section after the Introduction as "Literature review: Definitions and theoretical framework" (line 62).
  4. As you proposed, most of the references you mentioned are now included in the study. It definitely makes it more comprehensive. Thank you for specifying them.
  5. As per your request, a new section was added after the Literature Review section, titled "Materials and Methods", to address the concerns you mentioned about the paper's methodology and design. This is highlighted in lines 274-293. It definitely adds value and clarity to the paper.
  6. In response to your last request, a section was added before the Conclusion, titled "Results Discussion", which provides a comparative analysis of results, articulate the novelty introduced by the article, briefly discuss the potential implications of the research, and outline the limitations of the study, and potential opportunities for research. This was done in lines 651-693. Much appreciation for that suggestion because it surely adds value (in terms of impact) and context to the paper that enriches its scholarly contribution.

Kind regards,

Round 2

Reviewer 2 Report

Comments and Suggestions for Authors

Thank you!

I was expecting more!!!!

Author Response

Dear Reviewer

Thank you for taking the time to review the manuscript in a second round. Please note that I have made some significant improvements to the paper. It is highlighted in green. I am sure the Editor will send you the latest version of the manuscript. I trust that it will now meet your expectations.

Kind regards

Reviewer 3 Report

Comments and Suggestions for Authors

After reviewing the article in the second round, the author has made improvements. However, in my judgment, the article is not yet ready for acceptance and publication. The following enhancements need to be made:

The literary review conducted thus far remains weak, and it is likely the primary contribution of the paper. However, it is not yet comprehensive. The author should enhance this aspect to strengthen the overall quality of the article.

For this particular type of article, an extensive literary review is imperative. A comprehensive review of the existing literature not only serves as the foundation for the research but also plays a crucial role in establishing the significance and context of the study. It provides readers with a thorough understanding of the existing knowledge landscape, identifies gaps or areas where the current research contributes, and sets the stage for the research question or hypothesis.

An in-depth literary review demonstrates the author's familiarity with the subject matter and showcases their ability to critically analyze and synthesize relevant literature. It helps in positioning the study within the broader academic discourse, showing the evolution of ideas, methodologies, and findings in the field. Moreover, a robust literary review aids in building credibility and trust among readers, as it reflects the author's commitment to scholarly engagement and the depth of their research.

Given the pivotal role of the literary review in shaping the intellectual framework of the article, it is crucial for the author to allocate sufficient attention and effort to this section. By expanding and strengthening the literary review, the author can not only enhance the overall quality of the paper but also ensure that the research is firmly grounded in the existing body of knowledge, contributing meaningfully to the academic conversation. Therefore, it is strongly recommended that the author focuses on refining and expanding the literary review to meet the necessary standards for publication. For example: Alonso, S. L. N. (2023). Can Central Bank Digital Currencies be green and sustainable? Green Finance, 5(4), 603–623. https://doi.org/10.3934/gf.2023023 (lines 447-450) or also  Rahman, S., Moral, I. H., Hassan, M., Hossain, G. S., & Perveen, R. (2022). A systematic review of green finance in the banking industry: Perspectives from a developing country. Green Finance, 4(3), 347–363. https://doi.org/10.3934/gf.2022017 (lines 442-445) or lines 143-145 for example Martin, V. (2023). Green finance: Regulation and instruments1. Journal of Central Banking Theory and Practice, 12(2), 185–209. https://doi.org/10.2478/jcbtp-2023-0019 among others. Please improve literature review.

Another crucial aspect that the author has attempted to improve but has not fully addressed is the lack of a clear methodology.

The described methodology appears to be geared towards a theoretical literature review to explore the synergy between green finance and economic inclusion. The study aims to investigate how this combination can expedite the achievement of Sustainable Development Goals (SDGs) and enhance economic sustainability. The analysis involves examining, evaluating, and synthesizing existing theoretical frameworks, concepts, and applications relevant to green finance and economic inclusion. However, there are areas that could be improved:

  1. Inclusion Criteria: Clearly detail the inclusion criteria used for selecting relevant studies and literature. This should include publication years, types of sources (e.g., peer-reviewed journals), languages, and any other relevant criteria to ensure the integrity and relevance of the review.
  2. Previous Studies and Replication Methodology: The text does not mention the existence of previous studies using the same methodology, and there is no discussion of whether the author replicates or adapts any existing methods.
    • Suggestion: Include a brief literature review to identify if there are previous studies addressing similar topics using comparable methods. Explicitly state whether the study replicates or adapts existing approaches.
  3. Qualitative and Narrative Approach: Justify in more detail why this qualitative approach was chosen over a quantitative one. Provide a stronger rationale for the methodological choice to enhance the study's foundation.
  4. Pillars of Economic Inclusion: Suggestion: Provide details on these pillars of economic inclusion so that readers can better understand how studies and literature are being assessed against these criteria.
  5. Sources and Literature Search: Detail how the literature search was conducted, including key terms used and how sources were selected. This information can enhance transparency and reproducibility.

In summary, while the methodology outlines a theoretical and qualitative approach, providing greater clarity and detail on inclusion criteria, replication methodology, rationale for the chosen approach, pillars of economic inclusion, and the literature search strategy would strengthen the validity and replicability of the study.

Author Response

Dear Reviewer

Thank you for taking the time to review the manuscript in a second round. Please find the responses below and the corresponding revisions/corrections highlighted in green in the re-submitted paper, which you should receive from the Editor. 

  1. The literature review has been improved significantly, taking all your suggestions and recommendations into consideration. This is observable in the latest version of the manuscript between lines 273-397.
  2. The methodology has also been improved substantially. Thank you for pointing out a number of areas in which it can be enhanced. This paragraphs that were added is from lines 419-482.

I trust that the literature review is now more comprehensive and that the methodology is clearer and better explained.

Round 3

Reviewer 3 Report

Comments and Suggestions for Authors

Dear Author(s),
I note that you have made the requested changes. The paper is ready for acceptance.

Yours sincerely,